# Current Insights in Prolactin Signaling and Ovulatory Function

**DOI:** 10.3390/ijms25041976

**Published:** 2024-02-06

**Authors:** Dariusz Szukiewicz

**Affiliations:** Department of Biophysics, Physiology & Pathophysiology, Faculty of Health Sciences, Medical University of Warsaw, 02-004 Warsaw, Poland; dariusz.szukiewicz@wum.edu.pl

**Keywords:** prolactin, prolactin receptor, prolactin signaling, hypothalamic–pituitary–gonadal (HPG) axis, ovulation, ovulatory disorders, hyperprolactinemia, tuberoinfundibular dopamine (TIDA) neurons, kisspeptin, gonadotropin-releasing hormone (GnRH) neurons

## Abstract

Prolactin (PRL) is a pleiotropic hormone released from lactotrophic cells of the anterior pituitary gland that also originates from extrapituitary sources and plays an important role in regulating lactation in mammals, as well as other actions. Acting in an endocrine and paracrine/autocrine manner, PRL regulates the hypothalamic–pituitary–ovarian axis, thus influencing the maturation of ovarian follicles and ovulation. This review provides a detailed discussion of the current knowledge on the role of PRL in the context of ovulation and ovulatory disorders, particularly with regard to hyperprolactinemia, which is one of the most common causes of infertility in women. Much attention has been given to the PRL structure and the PRL receptor (PRLR), as well as the diverse functions of PRLR signaling under normal and pathological conditions. The hormonal regulation of the menstrual cycle in connection with folliculogenesis and ovulation, as well as the current classifications of ovulation disorders, are also described. Finally, the state of knowledge regarding the importance of TIDA (tuberoinfundibular dopamine), KNDγ (kisspeptin/neurokinin B/dynorphin), and GnRH (gonadotropin-releasing hormone) neurons in PRL- and kisspeptin (KP)-dependent regulation of the hypothalamic–pituitary–gonadal (HPG) axis in women is reviewed. Based on this review, a rationale for influencing PRL signaling pathways in therapeutic activities accompanying ovulation disorders is presented.

## 1. Introduction

Reports on the lactogenic (pro-lactogenic) activity of extracts from the anterior pituitary gland of cows first appeared in 1928 in a publication by P. Stricker and F. Grueter from the laboratory of Bouin at the University of Strasbourg, France [1,2]. Stimulation of lactation in rabbits via the injection of pituitary extracts was also confirmed in pigeons by O. Riddle, R.W. Bates, and S.W. Dykshorn at the Cold Spring Harbor Laboratory on Long Island, NY, USA [3,4], based on observations of crop milk production. The latter team of authors made significant contributions to the identification, isolation, and purification of a compound that acts as a polypeptide hormone and a tetrahelical cytokine, which was referred to as prolactin (PRL) because of its role in lactation [5,6].

After many years of intensive research, it is now known that PRL has a wide range of other diverse functions in the body [7,8,9,10], influencing not only the reproductive system [9,10,11,12,13,14,15] but also growth and development [16], metabolism [17,18,19,20], electrolyte transport [21], the integumentary system [22,23], behavior [24], the immune system [25,26,27], and carcinogenesis [28,29,30]. PRL exerts these versatile actions by acting on its receptor (PRLR), which is an archetype member of the class I cytokine receptor family [31,32,33].

Additionally, advancements in precise bioassay methods have revealed that PRL is not exclusively produced by anterior pituitary lactotrophs but is also synthesized at multiple extrapituitary sites under the control of the hypothalamus. The multiple extrapituitary sites at which PRL is expressed and secreted in humans include the brain, endothelial cells, immune cells, decidua, and ovarian tissue [10,34]. Pituitary PRL expression has been shown to be dependent on the Pit-1 transcription factor, which is a member of the Pit-Oct-Unc (POU) homeodomain protein family, whereas the extrapituitary PRL gene promoter that allows for PRL expression in several nonpituitary cells and tissues has a dissimilar activity to the pituitary promoter, with Pit-1-independent activity and responsiveness to different regulators of gene expression [35].

The latter of the listed locations indicates that the human ovary is not only the target of endocrine PRL but also the site of PRL production [10,36,37]. In ovarian tissue, PRL is expressed in both the antral follicle, theca, and granulosa layers and is produced by granulosa cells [6,37,38,39]. Moreover, by acting locally as an autocrine or paracrine cytokine, this hormone is an important modulator of key processes such as follicular growth and development, angiogenesis, ovulation, and steroidogenesis [14,40]. Despite the lack of consensus regarding their detailed comprehensive classification, ovulatory disorders are among the leading causes of infertility [41,42]. In animals and humans, hypersecretion of PRL leads to inhibition of gonadotropin-releasing hormone (GnRH) secretion and to diminished GnRH receptor response to GnRH, together with a decline in luteinizing hormone (LH) pulse frequency and amplitude [43,44]. Therefore, abnormal PRL activity, both in the blood and at the level of ovarian tissue, can cause infertility due to the mechanism of anovulation or infrequent ovulation (oligo-ovulation) secondary to hormonal imbalances [44,45,46].

The aim of this review is to provide a comprehensive presentation of the current state of knowledge on the role of PRL in the process of ovulation and its disorders. Thus, a rationale for influencing PRL signaling pathways in therapeutic activities accompanying ovulation disorders is outlined.

## 2. Prolactin (PRL)

### 2.1. Structure

PRL is a globular protein hormone composed of a single-chain polypeptide with a molecular weight of 23 kDa that contains 199 amino acids in humans. There are three intramolecular disulfide bonds between the following six cysteine residues in the human PRL chain: Cys4-Cys11, Cys58-Cys174, and Cys191-Cys199 (Figure 1A) [6].

Analysis of the secondary structure of the PRL molecule showed that 50% of the length of the amino acid chain is arranged in α-helices, whereas the remaining part of the chain forms loops. According to the model of the tertiary structure of PRL, four long α-helices with four α-helical domains and two receptor binding sites are arranged in an antiparallel fashion (Figure 1B) [6,47].

Amino acid sequence homology with the human genome shows considerable species variation, ranging from 97% homology in primates to only 56% homology between primates and rodents [6,48]. In humans, PRL is structurally similar to growth hormone (GH) and human placental lactogen (hPL, also known as human chorionic somatotropin (hCS)). Together, these proteins form the “PRL/GH/hPL” family, which is characterized by a conserved helix bundle protein composition (Group I of the helix bundle protein hormones); consequently, PRL has similar biological activities to its representatives [49,50]. In humans, a single 10 kb gene encoding PRL is located on chromosome six and contains five exons and four introns [51]. The PRL gene transcription process is regulated by two independent promoter regions: the proximal region that controls pituitary-specific expression and the more upstream 5000-bp region that directs extrapituitary expression of PRL [52]. Preprolactin, which is a protein 2–3 kDa heavier than mature PRL, represents the initial product of PRL mRNA translation, from which PRL is produced after removal of the 28-amino acid signal peptide by proteolytic cleavage [6,8].

Alternative splicing of the primary transcript, as well as proteolytic cleavage and other posttranslational modifications (e.g., dimerization and polymerization, phosphorylation, glycosylation, sulfation, and deamidation) of the amino acid chain, cause human PRL to exist in numerous structural variants [48,53]. Therefore, in addition to the monomeric form of PRL, in the human body, PRL also occurs as a dimer (molecular weight (MW) of 48–56 kDa) composed of glycosylated monomers and in the form of macro-PRL or big–big PRL (MW > 150 kDa) composed of antigen–antibody complexes of monomeric PRL and immunoglobulin (most commonly, IgG) [54,55]. In a healthy person, monomeric PRL is the predominant form of circulating PRL and accounts for 85% of the total immunoreactive PRL, whereas dimeric PRL and macro-PRL each account for approximately 5–10% of the total PRL in the blood [55]. Importantly, routine tests cannot distinguish between the three forms of PRL, which may be important in diagnosing disorders of PRL production [56,57,58].

### 2.2. Prolactin Receptor (PRLR)

PRLRs belong to the lactogen/hematopoietic cytokine receptor superfamily. The same group of receptors as PRLRs, which consists of the simplest cytokine receptors, also includes the GH receptor, the interleukin-2 receptor (IL-2R), the erythropoietin receptor (EPOR), and the thrombopoietin receptor (TpoR, also known as MPL) [29,59]. Therefore, even with lower affinity, the PRLR can bind and be activated by hPL, GH, IL-2, EPOR, and TpoR [32]. PRLR lacks an intrinsic kinase domain but possesses a Janus kinase 2 (JAK2)-associated region. Therefore, the receptor chain is dependent on the associated kinases to transduce phosphorylation-based signaling cascades. Upon PRL stimulation, the PRLR transduces signals through the activation of JAK2, thus leading to the phosphorylation of JAK2 [60].

#### 2.2.1. PRLR Structure

The human PRLR gene is located at chromosomal region 5p14-p13. This gene exceeds 200 kb in length and contains 11 exons, including six noncoding exons 1 that are alternatively spliced to a common noncoding exon 2, as well as exons 3–10 that encode the full-length activating long form of the receptor [61,62]. As a transmembrane receptor, the PRLR is composed of three main regions: the extracellular, transmembrane (a single transmembrane-spanning domain), and intracellular (cytoplasmic) regions. The extracellular region comprises two fibronectin type III domains known as D1 and D2. The WSXWS motif in D2, which is a generic cytokine class I receptor, acts as a molecular switch during ligand-bound activation of the PRLR [63]. Conversely, proline-rich sequence-mediated JAK2 association with the PRLR occurs in the intracellular domain. This JAK2/PRLR interaction is essential, although it alone cannot induce signal transduction through the Box-1/Box-2 subdomains. Further interaction of Box-1 with JAK2 and SRC family kinases (e.g., Fyn and c-Src) is required [64,65].

The long form of the PRLR may regulate signal transduction by exploiting the properties of the PRLR intracellular domain, which is intrinsically highly unstructured/disordered and binds to negatively charged lipids of the inner plasma membrane through conserved motifs resembling immune receptor tyrosine-based activation motifs [66]. Such lipid association of the PRLR intracellular domain is not accompanied by induced folding and is independent of specific tyrosine phosphorylation [66,67].

Intermediate and various short forms result from alternative splicing, variable promoter usage, or other posttranscriptional (but unlikely) posttranslational modifications. Sequences from exon 11 are present only in the short forms of the receptor S1a and S1b and their respective variants [68,69]. Although many PRLR isoforms perform the same or similar biological roles, some of these isoforms have unique functions. Moreover, human PRLRs are located not only in the cell membrane but also in the cytosol (a soluble isoform) and, surprisingly, in cellular structures such as endosomes, the Golgi apparatus, and lysosomes (here, they are presumably a degraded form of the receptor) [62,66,70]. All of the soluble forms of the PRLR lack transmembrane and cytoplasmic regions [71,72]. The most common isoforms of the human PRLR are included in Figure 2.

#### 2.2.2. PRLR Signaling

PRLRs are expressed in a wide but varied range of human cells and tissues throughout an individual’s lifetime, starting from the preimplantation stage of the embryo and continuing further during intrauterine embryogenesis [7,73]. The binding of a PRL molecule produced in the pituitary or from extrapituitary sources to the extracellular ligand-binding domain of the PRLR initiates the signaling pathway within a systemic or local range, respectively. This also applies (to varying degrees and depending on the specific physiological or pathological conditions) to partial agonists of the PRLR (e.g., hPL, GH, and IL-2) [32]. Studies have also shown that the binding affinity of human PRLR for nonhuman PRL is lower than that for human PRL [74]. The signaling pathways associated with the long form of the PRLR are the most comprehensively characterized to date [70].

In canonical signaling, ligand binding to the PRLR results in stimulation of the tyrosine kinase activity of JAK2; moreover, with the exception of the short isoform of the receptor, multiple tyrosine residue phosphorylation within the PRLR occurs in the following ways. ❶ Subsequent activation of signal transducer and activator of transcription (STAT) proteins, particularly STAT5 (JAK2/STAT5 signaling), thus enforcing further downstream signaling [75,76]. Next, the tyrosine-phosphorylated STAT complex dissociates from the receptor, dimerizes, and translocates into the nucleus, where it binds to the promoters of target genes. Although the JAK/STAT pathway is considered one of the major downstream pathways for cytokine receptor signaling, PRL also activates ❷ the Ras kinase/Raf kinase/mitogen-activated protein kinase/extracellular signal-regulated kinase ½ (Ras/Raf/MAPK/ERK1/2) pathway [77,78] and ❸ the phosphoinositide 3-kinase/protein kinase B/mammalian target of rapamycin (PI3-Kinase/AKT/mTOR) [78,79] downstream signaling pathway. In addition, in the absence of a physical association with STAT5a, ❹ signaling through the transcription factors RUSH-1α, RUSH-1β, and SWI/SNF-related matrix-associated actin-dependent regulator of chromatin subfamily A, member 3 (RUSH/SMARCA3), have been identified (Figure 3) [80].

This diversity of signaling pathways corresponds to the numerous functions of PRL, which are associated with both physiological states and play important roles in the pathomechanism of diseases, especially endocrine disorders [7,8,10]. Moreover, these key PRL-mediated signaling pathways are integrated. The differences in the functions of PRLR according to the pleiotropic nature of PRL and other PRLR agonists are presented under selected physiological and pathological conditions in Figure 4. A detailed discussion of these activities, apart from those directly related to ovulation, is beyond the scope of this review.

## 3. Ovulation

In women, ovulation is a phase in the menstrual cycle when the rupture of the dominant follicle within one of the ovaries leads to the release of the egg (the secondary oocyte or an oocyte arrested in meiosis II at the stage of metaphase II), which enters the lumen of the fallopian tube and creates a chance for fertilization in the presence of fertile sperm [15,114]. Ovulation occurs around day 14 of a 28-day menstrual cycle (or generally at approximately 2 weeks before the onset of menstruation) in women of reproductive age (roughly from ages 12 to 49 years). Typically, human ovaries produce a single dominant follicle that is approximately 25 mm in diameter and contains approximately 50 million granulosa cells and 7 mL of follicular fluid at the time of ovulation. For ovulation to occur, the ovarian follicle (or a Graafian follicle), which becomes the dominant follicle as a result of a highly selective process, must undergo all stages of folliculogenesis (recruitment, selection, and ovulation) within a strictly defined time frame (Figure 5) [115,116,117,118,119,120,121,122,123,124], including those related to gene expression (Table 1).

A pool of ovarian follicles, numbering approx. 150,00–200,000 (in one ovary) during a woman’s puberty, is subject to recruitment and selection processes leading to the development of the dominant Graafian follicle, which undergoes ovulation [125,126,127]. It is worth noting that while the cyclical, gonadotropin-dependent recruitment of the early antral follicles takes place over 14-day periods (follicular phase), reaching this stage through the earlier stages of folliculogenesis (gonadotropin-independent in the case of primordial follicles and gonadotropin-responsive in the case of secondary follicles) takes much longer. The initiation of the menstrual cycle during puberty and its subsequent course are determined by the variable, pulsatile secretion of gonadoliberin (GnRH–gonadotropin-releasing hormone) from the hypothalamus [128]. GnRH stimulates the production and secretion of gonadotropins (FSH, follicle-stimulating hormone, and LH, luteinizing hormone) in the anterior pituitary gland (adenohypophysis), which in turn stimulates the development of ovarian follicles and the production of ovarian steroids by thecal and granulosa cells (GCs): estrogen (predominantly estradiol—E_2_) and progesterone (P4). The negative feedback mechanism within the hypothalamic–pituitary–ovarian (HPO) axis plays a key role in regulating the levels of all hormones that control the menstrual cycle [129]. However, there is an exception, when rising estradiol (E2) in the middle of the menstrual cycle paradoxically “switches” from being inhibitory on GnRH secretion (“negative feedback”) to stimulating GnRH release (“positive feedback”), resulting in a surge in GnRH secretion and a downstream LH surge that triggers ovulation. The onset of the LH surge usually precedes ovulation by 36 h, whereas the peak serum level of LH occurs 10–12 h before ovulation [130]. Among the regulatory factors of the HPO axis’ negative feedback, the heterodimeric proteins produced by ovarian follicles inhibin A (αβA, InhA) and B (αβB, InhB) play a significant role. During the normal menstrual cycle, serum InhB levels are lowest in the early follicular phase, reach a mid-cycle peak coincident with a decrease in FSH concentration and the preovulatory LH surge, and decrease during the luteal phase. Conversely, InhA concentrations are low during the selection and development of a dominant follicle, increase rapidly during ovulation, and are maximal during the mid-luteal phase [131]. Changes in sex hormone concentrations are accompanied by cyclic changes in the endometrium (not shown in the figure), which enable implantation of the blastocyst after oocyte fertilization or repetition of the cycle with the separation and expulsion of the functional layer of the uterine mucosa (onset of menstruation) [132,133].

**Table 1 ijms-25-01976-t001:** Expression of selected genes in individual structures of ovarian follicles during various stages of folliculogenesis. All genes were localized by in situ hybridization [134,135,136,137].

Class/Stage of the Follicle	Expression of Selected Genes		
	*Granulosa Cells*	*Oocyte*	*Theca Cells*
Primordial	*3βHSD*, *ALK3*, *BMPRII*, *Erβ*, *KITLG*, *StAR*, *WTI*	*ALK3*, *ALK6*, *BMP6*, *BMPRII*, *C-kit*, *Erβ*, *GDF9*, *GJA4*, *TGFBR3*	–
Primary	*βB-activin*, *ActRIIB*, *ALK6*, *AMH*, *AMHRII*, *FSH-R*, *GJA1*, *IGFR1*	*BMP15*, *FIGα*	–
Small preantral	*ALK5*, *FSRP*, *FST*, *TGFBR3*	–	*ActRIIB*, *ALK3*, *ALK5*, *ALK6?*, *BMPRII*, *FSRP*, *IGFR1*, *TGF-β1*, *TGF-β2*, *TGFBR3*, *TGFBR11*
Large preantral	*AR*, *ERα*, *InhA*	–	*3βHSD*, *ARErβ*, *CYP17A1*, *IGF2*, *LHR*, *PR*, *SF1*, *StAR*

The mechanisms regulating the selection of antral ovarian follicles are poorly understood and are thought to rely on low estrogen levels, a decrease in follicle-stimulating hormone (FSH) levels, and FSH receptor (FSHR) expression on the surface of granulosa cells. It has been assumed that a follicle capable of maximum expression of FSHR is able to maintain growth and becomes dominant in a given cycle [138,139]. However, such assumptions have not been confirmed under in vitro conditions, wherein apoptosis of human granulosa cells (hGLCs) is induced by high doses of FSH or FSHR overexpression by hGLC lines stably transfected with human FSHR cDNA, whereas estrogens induce antiapoptotic signals via nuclear and a G protein-coupled estrogen receptor (GPER). Therefore, in vitro data suggest that antral follicle selection may be driven by underestimated, FSH-FSHR-dependent apoptotic signals due to transient maximization of FSHR expression and overload of cAMP signaling, thus prevailing on estrogen-dependent signals [140,141,142].

In humans, a selected follicle becomes dominant approximately one week before ovulation; specifically, this occurs as early as 5 to 7 days into the cycle, at a time when the follicular diameter is approximately 10 mm. Only in the dominant follicle can FSH be detected at this point in time, which is also accompanied by a significant level of estradiol in the follicular fluid [122,123].

The remaining follicles that are recruited in a given cycle undergo atresia; in mammals, this is mainly a result of programmed cell death (apoptosis). Therefore, follicular atresia allows for the regulation of the number of follicles growing within a given pool by the crosstalk between cell death and cell survival signals. Accordingly, the fate of the individual follicle (growth to ovulation or inhibited growth with subsequent atresia) is dependent on a precise balance in the expression of specific receptors (mainly for endocrine hormones and gonadotropins) and the action of intraovarian factors (such as gonadal steroids, cytokines, and growth factors) regulating follicular cell proliferation, growth, and differentiation, as well as those promoting apoptosis. It is estimated that in each ovulatory cycle, the average ovary loses 1000 follicles during the process of selecting a dominant follicle that will be released. Therefore, the total pool of follicles must be sufficiently large [143,144].

Indeed, the fetal ovary at the twentieth week of gestation contains 6–7 million oocytes, which results from the mitosis of approximately 500 to 1300 primordial germ cells; this theoretically represents the entire cohort of oocytes capable of participating in reproduction during a woman’s lifetime [145]. These germ cells subsequently begin meiosis and arrest at meiotic prophase 1, wherein they form germ cell cysts and, ultimately, primordial follicles. Consisting of an oocyte surrounded by a single layer of cuboidal granulosa cells that initiates follicle development, primordial follicles are the first class of follicles that develop in mammalian ovaries. The process of follicular recruitment begins soon after this time point and proceeds continuously in both gonadotropin-independent (before puberty) and gonadotropin-dependent (after puberty) manners. Consequently, the number of follicles in the ovaries decreases to approximately 500,000 at birth, 150,000 at puberty, and 1000 at menopause [146]. In simple terms, this means that of the entire pool of ovarian follicles present on the day a woman is born only 0.1% will ovulate, while 99.9% will be degraded in the process of follicular atresia [147,148]. The continuity of folliculogenesis means that an active ovary always contains follicles in various stages of development [149].

The luteal phase begins immediately after ovulation. This phase is usually 14 days long in most women. After the oocyte with cumulus cells is released from the ovulating follicle, the remaining granulosa cells continue to enlarge, become vacuolated, and begin to accumulate a yellow pigment known as lutein. Luteinized granulosa cells with newly formed theca-lutein cells and surrounding stroma give rise to a new structure known as the corpus luteum. The primary hormone produced by this transient endocrine organ is progesterone; however, it also produces inhibin A and estradiol. The primary function of the corpus luteum is to prepare the estrogen-primed endometrium for implantation of the fertilized ovum. This transformation of the endometrium is known as decidualization. In contrast to most mammals, decidualization of the human endometrium does not require embryo implantation. Instead, this process is driven by the postovulatory rise in progesterone levels and increasing local cAMP production [150]. Eight or nine days after ovulation, approximately around the time of expected implantation, peak vascularization is observed within the corpus luteum. This time period also corresponds to peak serum levels of progesterone and estradiol [118]. The lifespan of the corpus luteum depends upon continued LH support. Corpus luteum function declines by the end of the luteal phase unless human chorionic gonadotropin (hCG) is produced during a pregnancy. If pregnancy does not occur, the corpus luteum undergoes luteolysis under the influence of estradiol and prostaglandins, during which a scar tissue known as the corpus albicans is formed [118]. In response to decreasing progesterone levels, spontaneous decidualization causes menstrual shedding and cyclic regeneration of the endometrium.

### 3.1. Hormonal Regulation of Ovulation 

Occurring during the course of the menstrual cycle, the morphological changes in the ovaries leading to ovulation and the formation of the corpus luteum, as well as functional changes in the endometrium that enable blastocyst implantation in the event of fertilization of the ovum, are primarily regulated by hormones [151,152,153]. The pulsatile secretion of gonadotrophin-releasing hormone (GnRH) from the hypothalamus (under the control of the kisspeptin-neurokinin B-dynorphin (KNDγ) pathway) stimulates the anterior pituitary to secrete a follicle-stimulating hormone (FSH) and a luteinizing hormone (LH), which correspondingly stimulate the development of ovarian follicles and the production of the ovarian steroids estrogen (predominantly estradiol) and progesterone (P4) [153,154,155,156]. A negative feedback mechanism within the hypothalamic–pituitary–ovarian (HPO) axis is crucial for the control and regulation of the menstrual cycle, particularly during the following three phases: follicular (which begins with menstrual bleeding), ovulatory, and luteal phases. However, there is an exception in females, wherein rising estradiol (E2) during the middle of the menstrual (or estrous) cycle paradoxically “switches” from being inhibitory on GnRH secretion (“negative feedback”) to stimulating GnRH release (“positive feedback”), thus resulting in a surge in GnRH secretion and a downstream LH surge that triggers ovulation (Figure 6B) [157,158]. Additionally, the granulosa cells of the growing follicle secrete a variety of peptides that may play an autocrine/paracrine role in stimulating follicles (activins), thus inhibiting the development of adjacent follicles (e.g., inhibins and follistatin) or suppressing/unsuppressing the biosynthesis of FSH (inhibins vs. activins) [131,159,160,161]. The luteal phase of the cycle is relatively constant in all women, with a duration of 14 days. The variability of cycle length is usually derived from varying lengths of the follicular phase of the cycle, which can range from 10 to 16 days [118]. 

#### The General Characteristics of the Main Endocrine Hormones Involved in the Menstrual Cycle and Ovulation (See also Table 2 for the Schematic Structure)

Please note that the normal values/concentration ranges for individual hormones listed below often vary according to literature data. Current reference values from recognized clinical centers for the treatment of menstrual cycle disorders and infertility are provided here [165,166].

**Table 2 ijms-25-01976-t002:** Schematic structure of the main hormones involved in the menstrual cycle and ovulation.

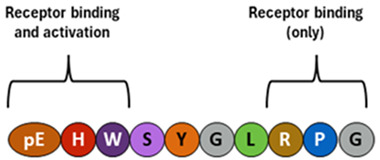 **Gonadotropin-releasing hormone (GnRH)**	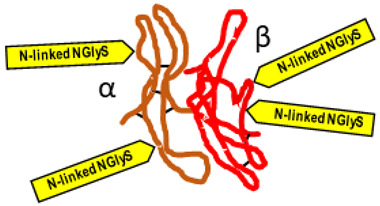 **Follicle-stimulating hormone (FSH)**	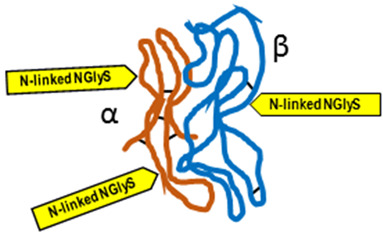 **Luteinizing hormone (LH)**
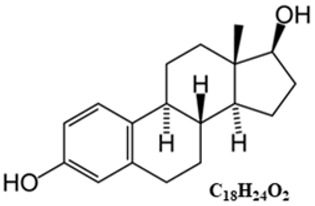 **Estradiol (E_2_)**	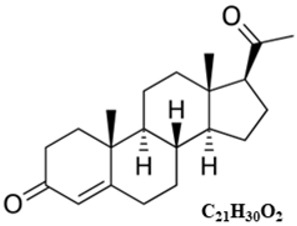 **Progesterone (P4)**	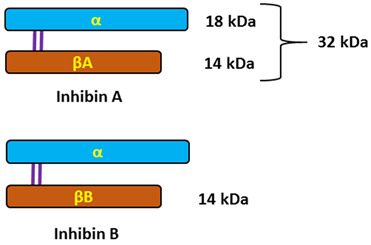 **Inhibins**


**Gonadotropin-releasing hormone (GnRH).**


*Source.* Neurons of hypothalamus.

*Typical structure or chemical formula.* GnRH is a decapeptide with the presence of the amino-terminal pyroglutamic acid (pE, a cyclic non-proteinogenic amino acid, containing a γ-lactam ring, that is produced from glutamine or glutamic acid by deamidation or dehydration, respectively) and the amidated carboxy terminus. In all three forms of GnRH (GnRH1, GnRH2, and GnRH3), both N-terminal and C-terminal are conserved, which allows for effective binding to their receptors [115,167,168].

*General description.* The master hormone regulates reproductive activity by stimulating the release of gonadotropins and, consequently, stimulating the production of sex hormones in the gonads. This hormone ultimately regulates puberty onset, sexual development, and ovulatory cycles in females [115,116].

*Level changes during the menstrual cycle and ovulation.* The frequency and amplitude of hypothalamic GnRH pulses determine the relative proportions of pituitary secretion of FSH and LH. In the normal menstrual cycle, GnRH pulse frequency increases from about 90–100 min to about every 60 min through a follicular phase. A gradual increase in GnRH pulse frequency facilitates LH secretion culminating in an ovulatory LH surge [116,169]. Earlier, a switch from estradiol negative to positive feedback initiates the GnRH influx, affecting LH release. In contrast, increased progesterone secretion in a luteal phase results in an increase in FSH synthesis as a result of less frequent hypothalamic GnRH secretion (approximately one pulse every 3–5 h) through mechanisms involving opioid receptors [170,171,172,173] and possibly other factors such as kisspeptin [174].

*Concentration range.* Approx., 0.1–2.0 pg/mL; the basal levels of GnRH do not change significantly before, during, and after the LH surges and show fluctuations (pulsations) between a small range of 0.1 and 2.0 pg/mL [175].


**Follicle-stimulating hormone (FSH)**


*Source.* Anterior pituitary.

*Typical structure or chemical formula.* FSH is a 35.5 kDa glycoprotein heterodimer consisting of two non-covalently linked polypeptide subunits: the alpha subunit (92 aa), which is common to all glycoprotein hormones, and a unique beta subunit (111 aa), providing specificity of action. The FSH structure is similar to those of luteinizing hormone (LH), thyroid-stimulating hormone (TSH), and human chorionic gonadotropin (hCG) [176]. Each subunit contains two asparagine-linked N-glycosylation sites (N-linked NGlyS): N52 and N78 within the α subunit and N7 and N24 within the β subunit [177]. Based on the number of occupied N-glycosylation sites, four human FSH glycoforms have been identified: hFSH24, which possesses N-glycans at all four sites; hFSH21, which lacks the βAsn24 glycan; hFSH18, which lacks the βAsn7 glycan; and hFSH15, which lacks both FSHβ N-glycans. The two most abundant human FSH glycoforms are FSH24 and FSH21 [178].

*General description.* Produced in response to GnRH, FSH plays a central role in reproduction. In females, FSH stimulates antrum formation in secondary follicles, growth, and maturation in antral follicles, and it prepares the latter for ovulation in response to the LH surge [117]. FSH stimulates granulosa cells in the ovarian follicles to synthesize aromatase, which converts androgens produced by the thecal cells to estradiol [E2]. In the presence of estradiol, FSH stimulates the formation of LH receptors on granulosa cells allowing for the secretion of small quantities of progesterone and 17-hydroxyprogesterone (17-OHP), which may exert a positive feedback on the estrogen-primed pituitary to augment LH release [118,179,180].

*Level changes during the menstrual cycle and ovulation.* Declining steroid production by the corpus luteum and the dramatic fall of inhibin A allows for FSH to rise during the last few days of the menstrual cycle. Another influential factor on the FSH level in the late luteal phase is related to an increase in GnRH pulsatile secretion secondary to a decline in both E2 progesterone (P4) levels. This elevation in FSH allows for the recruitment of a cohort of ovarian follicles in each ovary, one of which is destined to ovulate during the next menstrual cycle [117]. Increasing FSH levels during the late luteal phase leads to an increase in the number of FSH receptors (FSHRs) and ultimately to an increase in estradiol secretion by granulosa cells. It is important to note that the increase in FSHR numbers is due to an increase in the population of granulosa cells and not due to an increase in the concentration of FSHR per granulosa cell. Each granulosa cell has approximately 1500 FSHRs by the secondary stage of follicular development and FSHR numbers remain relatively constant for the remainder of development [181]. Once menses ensue, FSH levels begin to decline due to the negative feedback of estrogen and the negative effects of inhibin B produced by the developing follicle (Figure 5) [118].

*Concentration range.* Early follicular 3–10 (IU/L); mid-cycle peak 4–25; pregnancy < 1.


**Luteinizing hormone (LH)**


*Source.* Anterior pituitary.

*Typical structure or chemical formula.* Human LH is a 29.0 kDa glycoprotein heterodimer consisting of two non-covalently linked polypeptide subunits: the alpha subunit (92 aa), which is common to all glycoprotein hormones (including FSH, hCG, and TSH), and a unique beta subunit (120 aa) that confers its specific biological action. The β-subunit of LH contains an amino acid sequence that exhibits large homologies with that of the beta subunit of hCG, and both stimulate the same receptor. However, the hCG beta subunit contains an additional 24 amino acids, and the two hormones differ in the composition of their sugar moieties, which affects bioactivity and half-life. There are three asparagine-linked N-glycosylation sites (N-linked NGlyS): N52 and N78 within the α subunit and only one carbohydrate attachment site (N-linked NGlyS) within the β subunit [119,182].

*General description.* LH is essential to provide the androgen substrate for estrogen synthesis, which in turn contributes to oocyte maturation and may play a relevant role in optimizing fertilization and embryo quality [119,183]. LH helps to regulate the length and order of the menstrual cycle by playing roles in both ovulation and implantation of an egg in the uterus [180,184].

*Level changes during the menstrual cycle and ovulation.* Unlike FSH, LH concentration is low during the early follicular phase and begins to rise by the mid-follicular phase due to the positive feedback from the rising estrogen levels. For the positive feedback effect of LH release to occur, estradiol levels must be greater than 200 pg/mL for approximately 50 h in duration [118]. The midcycle gonadotropin (FSH, LH) surge marks the end of the follicular phase of the cycle and precedes actual rupture by as much as 36 h. During this surge, LH levels are highest about 10–12 h before ovulation and can reach 30 IU/L or higher. The LH surge stimulates luteinization of the granulosa cells and stimulates the synthesis of progesterone responsible for the midcycle FSH surge. Also, the LH surge stimulates the resumption of meiosis and the completion of reduction division in the oocyte with the release of the first polar body. LH and progesterone cause an increase in the activity of prostaglandins and proteolytic enzymes (e.g., collagenase and plasmin) involved in the mechanism of follicle wall rupture with release of the oocyte-cumulus of the granulosa cells complex [119,185]. After ovulation, the remaining theca and granulosa cells in the crater of the ovulated follicle organize into a progesterone-secreting corpus luteum and are active for about 2 weeks, after which it regresses in the absence of pregnancy (Figure 5).

*Concentration range.* Early follicular 1.5–9 (IU/L); mid-cycle peak (before ovulation) 6.2–35; weeks 3 and 4 of the menstrual cycle: 1–9.2; pregnancy < 2–9.


**Estrogens**


*Source.* Ovaries (ovarian follicles).

*Typical structure or chemical formula.* Chemically, estrogens are derivatives of cholesterol, an organic compound belonging to the steroid family. There are three major endogenous estrogens that have estrogenic hormonal activity: estrone (E1), estradiol (E2), and estriol (E3). Another type of estrogen called estetrol (E4) is produced only during pregnancy. Steroid sex hormone estradiol (E2; 17β-estradiol) is the most abundant and active estrogen in women of reproductive age (between the menarche and menopause) [186]. In females, estrogens are produced by locally expressed p450 aromatase from follicular androgens in the ovary and other estrogen-responsive tissues [187].

*General description.* Estrogen is one of the most impactful hormones in the body. Estrogen is responsible for the stimulation of secondary female characteristics (body composition, breast development, menstrual cycle, etc.). It can also impact other aspects of health such as control of metabolism (e.g., bone mineral density, cholesterol level), mood stability, immune system function, complexion, and carcinogenesis [188,189,190,191]. Estrogens are produced primarily by the ovaries. They are released by the follicles on the ovaries and are also secreted by the corpus luteum after the oocyte has been released from the follicle. During pregnancy, the placenta is an important source of estrogens [192].

*Level changes during the menstrual cycle and ovulation.* Estrogen (predominantly E2) levels rise and fall twice during the menstrual cycle. Estrogen levels rise during the mid-follicular phase and then drop precipitously after ovulation. This is followed by a secondary rise in estrogen levels during the mid-luteal phase with a decrease at the end of the menstrual cycle (Figure 5) [118].

*Concentration range.* Early follicular < 300 pmol/L; ovulatory surge 500–3000 pmol/L; luteal surge 100–1400 pm/L.


**Progesterone (P4)**


*Source.* Ovaries (corpus luteum).

*Typical structure or chemical formula.* In mammals, P4, like all other steroid hormones, is synthesized from pregnenolone, which itself is derived from cholesterol. P4 biosynthesis requires only two enzymatic steps; the conversion of cholesterol to pregnenolone, catalyzed by P450 side chain cleavage (P450scc) located on the inner mitochondrial membrane, and its subsequent conversion to progesterone, catalyzed by 3β-hydroxysteroid dehydrogenase (3β-HSD) present in the smooth endoplasmic reticulum (SER) [193].

*General description.* The synthesis of progesterone by the corpus luteum (luteinized granulosa and theca cells) is essential for the establishment and maintenance of early pregnancy (decidualization of the endometrium) [194]. In the following weeks of pregnancy (after the 10th week of pregnancy), the placenta becomes the main source of progesterone [195].

*Level changes during the menstrual cycle and ovulation.* P4 levels are low during the follicular phase, and they rise after ovulation during the luteal phase, with the possibility of further maintenance and increase of P4 biosynthesis as a result of stimulation with chorionic gonadotropin (hCG) after blastocyst implantation [120].

*Concentration range.* Follicular phase: <0.181 to 2.84 nmol/L; luteal phase: 5.82–75.9 nmol/L; detecting ovulation—measured on day 20–23 of a normal 28-day cycle: >25 nmol/L ovulation likely; 7–25 nmol/L—ovulation possible; 0–6 nmol/L—ovulation unlikely.


**Inhibins**


*Source.* Ovaries (ovarian follicles: granulosa and theca cells).

*Typical structure or chemical formula.* Inhibins are heterodimeric glycoproteins composed of an α-subunit and one of two β-subunits, forming inhibin A (αβA) and inhibin B (αβB). The respective subunits are linked via a single disulfide bond. The α-and β-subunits are produced as larger precursor proteins, prepro-α, pro-βA, and pro-βB, that include a signal peptide and pro-region, both of which are cleaved to form the mature α or β-subunit [121,196,197]. Inhibins, similarly to activins that have opposing effects on FSH secretion, belong to the transforming growth factor-beta (TGF-β) superfamily of multifunctional cytokines [122].

*General description.* The activins and inhibins are among the 33 members of the TGF-β superfamily and were first described as regulators of follicle-stimulating hormone (FSH) secretion and erythropoiesis. It is now known that these cytokines participate in the regulation of various processes, ranging from the early stages of embryonic development to highly specialized functions in terminally differentiated cells and tissues that maintain homeostasis [122,198,199,200]. The biological effects of TGF-β family proteins, including inhibins, are contextual, and even the same cell type may show different or opposite responses to the ligand under different biological contexts. It is, therefore, possible that the heterogeneity could contribute to different responsiveness to TGF-β family members [201]. In female individuals, inhibin A is primarily produced by the dominant follicle and corpus luteum, whereas inhibin B is predominantly produced by small developing follicles. The negative feedback control of pituitary FSH secretion has been the most recognized physiological role of inhibins. They selectively suppress the secretion of pituitary follicle-stimulating hormone (FSH) and have local paracrine actions in the gonads [118,123,202].

*Level changes during the menstrual cycle and ovulation.* Serum inhibin A and B levels fluctuate during the menstrual cycle. Inhibin B dominates the follicular phase of the cycle, while Inhibin A dominates the luteal phase. This means that serum inhibin B levels increase early in the follicular phase to reach a peak coincident with the onset of the decline in FSH levels at the mid-follicular phase, whereas inhibin A levels are low in the early follicular phase and rise at ovulation to maximum levels in the mid-luteal phase of the menstrual cycle (Figure 5). Once menses ensue, FSH levels begin to decline due to the negative feedback of estrogen and the negative effects of inhibin B produced by the developing follicle [124,203].

*Concentration range.* 2–80 pg/mL: 2–10 pg/mL in the follicular phase, 40–80 pg/mL in the luteal phase.

Importantly, all ovarian hormones that regulate the course of the menstrual cycle are synthesized under precisely defined conditions and time frames and act in an autocrine and paracrine manner. A comprehensive understanding of the mechanisms of influence at this level in physiological states and disorders (e.g., ovulation) requires further research [140,204,205,206].

### 3.2. Mechanism of Follicle Rupture during Ovulation

The initial hypothesis that the rupture of the wall of the ovulating follicle occurs as a result of an increase in intrafollicular pressure has not been confirmed by different studies. Despite a significant and rapid increase in the volume of follicular fluid before ovulation, which is mainly due to the inflow of fluid from the extravascular space, no significant changes in intrafollicular pressure have been detected at that time [207].

Therefore, it is important that the preovulatory surge in gonadotropins (primarily LH) triggers a series of changes in the wall of the dominant follicle, the enzymatic degradation of which leads to rupture and determines the extrusion of the oocyte. Therefore, the assessment of the molecular mechanisms of follicle wall rupture during ovulation has focused on creating a model of interconnected signaling networks that initiate a proteolytic enzyme cascade that has a local effect. Many mediators of these LH-induced signaling cascades are associated with inflammation, including the production of mediators of the inflammatory response by granulosa and theca cells, such as steroids, prostaglandins, chemokines, and proteolytic enzymes (e.g., collagenase and plasmin) [185]. It has been demonstrated that granulosa cells in vivo produce increasing amounts of plasminogen activators as ovulation approaches and that the enzyme is solely produced by cells obtained from follicles destined to ovulate [208]. The production of a plasminogen activator is regulated by gonadotropins [208,209].

The concentrations of prostaglandins E and F and hydroxyeicosatetraenoic acid (HETE) reach a peak in the follicular fluid just prior to ovulation [210,211].

Prostaglandins may stimulate both proteolytic enzymes and oocyte release by inducing smooth muscle fiber contractions within the ovary, whereas HETE may stimulate angiogenesis and hyperemia [118,211]. The point at which the dominant follicle is closest to the ovarian surface where digestion and subsequent rupture of the wall occur is known as the stigma.

In the context of the obvious similarities between ovulation and the inflammatory response, it is worth noting that at the level of ovarian follicles, PRL (which is also locally produced in the ovarian tissue) acts both as a hormone (endocrine and paracrine) and as a proinflammatory cytokine [6,212]. Thus, under certain conditions, the expression of PRLRs plays both pro- and anti-gonadal roles in the regulation of ovarian functions, including ovulation. For example, PRL may inhibit ovulation by stimulating granulosa cell apoptosis or promote ovulation by triggering a cascade of proteolytic processes in the wall of the ovarian follicle. However, neither the pattern of expression of PRLRs nor its regulatory mechanisms during follicle development have been clearly defined [213,214].

### 3.3. Ovulatory Disorders

The term “ovulatory disorders” describes a set of abnormalities that occur continuously or intermittently, thus resulting in the absence of ovulation (anovulation) or its infrequent, irregular occurrence (oligoovulation) [215]. During oligoovulation, the menstrual cycle is typically longer than normal (21 to 35 days), thus resulting in eight or fewer menstrual periods per year. In addition to abnormal uterine bleeding, ovulatory disorders may disappear in some cases. Ovulatory disorders, which typically result from episodic or chronic dysfunction of the hypothalamic–pituitary–ovarian (HPO) axis, represent a major cause of infertility, as the lack of an optimal quality oocyte that is regularly released every month significantly reduces the chances of fertilization. Anovulation and oligoovulation are estimated to constitute 25% of the known causes of female infertility [162,216].

Although there are numerous known causes and contributors to ovulatory disorders, the entire spectrum of mechanisms of pathogenesis remains to be fully elucidated. Ovulatory disorders are often observed due to coexisting endocrinopathy, neoplasms, or emotional states requiring psychological/psychiatric assistance and as a consequence of the use of medications. Therefore, optimally effective research, teaching, and clinical management of ovulatory disorders must be based on a generally accepted and possibly simple classification. Intensive activities that have been recently conducted by international bodies (International Federation of Gynecology and Obstetrics, or FIGO) have resulted in a consensus and the presentation of a proposal for a new classification; according to its authors, this proposal may replace the existing ones, especially the widely used World Health Organization (WHO) classification of ovulatory dysfunction [163,164,216]. Both classifications are summarized in Figure 6.

The **WHO classification** clearly takes into account the states of hyperprolactinemia in groups V and VI (highlighted on a yellow background).

In the FIGO classification, the effect of PRL on ovulation is not directly separated at various levels (hypothalamic, pituitary, or ovarian) of physiological and pathological (PCOS) regulation; in contrast, the WHO classification clearly considers the states of hyperprolactinemia in groups V and VI.

According to various estimates, polycystic ovary syndrome (PCOS), which is the most common cause of oligoovulation and anovulation (anovulatory infertility), may affect 5 to even 20% of women of reproductive age [217,218,219,220]. In PCOS, abnormal hormone levels prevent ovarian follicles from growing and maturing to ovulation. As a result, immature follicles accumulate in the ovaries, thus resulting in a typical picture of polycystic morphology via ultrasound examination [221]. The pathophysiology of PCOS is heterogeneous and associated with interactions between reproductive function disorders and altered metabolism [222]. Hyperandrogenism and insulin resistance are part of a vicious disease cycle that exacerbates the course of PCOS and is also affected by dysfunction of the hypothalamic–pituitary–ovarian (HPO) axis [223]. Notably, in 30% of PCOS patients, an increase in the serum PRL concentration can be detected in both the follicular and luteal phases of the menstrual cycle [224,225]. It is well-documented that hyperprolactinemia may limit the number of growing ovarian follicles and inhibit ovulation [226,227].

#### 3.3.1. PRL and Ovulatory Disorders

The average basal level of PRL in women is 13 ng/mL (277 mIU/L), whereas the upper normal limit of serum PRL in most laboratories is 15 to 20 ng/mL (320 to 425 mIU/L) [228]. Elevated PRL (hyperprolactinemia) is the most common endocrine disorder of the hypothalamic–pituitary axis. Clinical data clearly indicate an association between hyperprolactinemia and infertility, the main cause of which is ovulatory disorders due to suppression of the HPO axis via inhibition of a pulsatile gonadotropin-releasing hormone [46].

The vast majority of causes of hyperprolactinemia can be classified as physiological, pathological, or drug-induced (iatrogenic) [44,228,229]. However, despite continuous progress in pathophysiology and diagnostic methods, cases of idiopathic hyperprolactinemia of unknown or no clear cause, as well as hyperprolactinemia, are misdiagnosed due to the use of inadequate analytical techniques [230,231]. Table 3 categorizes the most important causes of hyperprolactinemia, which may result in ovulatory disorders, and discusses the known or probable underlying mechanisms. Naturally, as the table shows, increased PRL concentration at a given moment may result from several long-term or temporary factors.

Hypothalamic dopaminergic cells, which are known as tuberoinfundibular dopamine (TIDA) neurons, are located in the arcuate nucleus and send axonal projections to the median eminence. Dopamine is tonically (i.e., continuously) secreted by these dopaminergic neurons to ensure an appropriate level of inhibition of PRL release from pituitary lactotrophs [8]. Therefore, increased PRL secretion under both physiological and pathological conditions mostly results from the removal of dopamine-inhibiting pathways. Hence, most prolactin-releasing factors (PRFs) act indirectly through the deactivation of the TIDA system, although direct effects on lactotrophs and other mechanisms of action (which have not been completely clarified) are certainly present [270,271,272]. Although PRL is well known to stimulate hypothalamic dopamine secretion, thereby exerting negative feedback regulation on its own release, autocrine or paracrine actions of PRL on lactotroph cells have also been suggested [273].

The most common cause of hyperprolactinemia is prolactinoma, which involves a benign (noncancerous) prolactin-releasing tumor, the main prevalence of which is estimated to be approximately 30 per 100,000 people, with a peak incidence occurring at ages 25 to 34 years [274]. Moreover, even in approximately 40% of nonfunctioning pituitary adenomas (NFPAs), benign adenohypophyseal tumors are not associated with clinical evidence of hormonal hypersecretion, and hyperprolactinemia occurs as a result of pituitary lactotroph disinhibition (the stalk effect) [275]. Interestingly, most patients with increased intrasellar pressure caused by large pituitary tumors do not exhibit hyperprolactinemia. Lactotroph insufficiency accompanying pituitary failure may be responsible for 87% of cases of normoprolactinemia among patients with pituitary tumor sizes > 20 mm [276]. Similarly, cases of hypoprolactinemia in the context of ovulatory disorders should be considered accompanying damage to the entire glandular part of the pituitary gland (e.g., as a result of puerperal ischemia and necrosis in Sheehan’s syndrome); therefore, it can be combined with gonadotropin deficiencies [277].

As shown in Table 3, a large group of medications can cause an increase in PRL levels, thus impairing dopaminergic activity in the central nervous system (CNS) in various ways, including by acting as a D2 receptor antagonist, a false neurotransmitter, an H2 receptor antagonist, a Ca^2+^ channel blocker, a selective 5-HT reuptake inhibitor (SSRI) or an inhibitor of the vesicular monoamine transporter type 2 (VMAT2) [278,279]. Iatrogenic causes of hyperprolactinemia may be a clinical problem in which the incidence is underestimated, especially in patients who are not taking antipsychotics or antidepressants [104,258,280]. For example, hyperprolactinemia caused by treatment of pregnancy-induced hypertension (PIH) with methyldopa may be associated with an increased incidence of postpartum depression and maternity blues (also known as baby blues) [281].

Prolactinomas are treated with surgery or DA agonists (e.g., bromocriptine, lisuride, quinagolide, and cabergoline) depending on the adenoma size, clinical factors, and patient preference. In microadenomas, patient preference for observation or hormonal replacement therapy (HRT) can also be considered depending on menopausal and gonadal status [282,283]. It is estimated that treating hyperprolactinemia with DA agonists leads to the normalization of PRL levels and the return of ovulatory cycles in approximately 80% of patients. In treatment-resistant patients, the DA agonist is recommended to be changed [283]. In hyperprolactinemia induced by drugs that cannot be discontinued due to the underlying disease, DA administration may be pointless or even dangerous. Once a pituitary adenoma has been ruled out, the use of sex steroids (HRTs) is recommended in this situation (if necessary) to prevent osteoporosis [283,284].

Macroprolactinemia, which is defined as the quantitative predominance of an isoform of a greater molecular weight than PRL known as macroprolactin (big–big PRL), is a common cause of hyperprolactinemia. According to various data, among patients with hyperprolactinemia, 10–46% were diagnosed with macroprolactinemia [57,58,285]. However, due to the fact that it is usually composed of a PRL monomer and an IgG molecule that has a prolonged clearance rate similar to that of immunoglobulins, macroprolactin is characterized by limited bioavailability (as confined to the vascular system) and much lower bioactivity than PRL. Therefore, the predominance of macroprolactin, which is the main molecular form of PRL in the serum of patients with a normal concentration of monomeric PRL, is associated with no symptoms or a mild course of hyperprolactinemia [52,55,229,286]. 

Given the high prevalence of macroprolactinemia among women with elevated PRL levels and the difference in the management of patients with macroprolactinemia compared to true monomeric hyperprolactinemia, all patients with persistently elevated PRL levels, especially asymptomatic patients, should be screened with tests appropriate for the diagnosis of macroprolactinemia [54,287,288,289].

##### PRL and the Release of Gonadotropins

A sufficiently high PRL concentration suppresses hypothalamic gonadotropin-releasing hormone (GnRH, also known as gonadoliberin), which is part of a family of peptides that play pivotal roles in reproduction by stimulating the synthesis and secretion of LH and FSH from the adenohypophysis [82].

When considering the physiological activities of PRL, other than those related to lactation, it can be assumed that a certain level of PRL circulating in the blood, as well as what is produced locally in reproductive tissues, may be necessary for optimal oocyte development, formation and maintenance of the corpus luteum, as well as blastocyst implantation, steroidogenesis in early pregnancy, and modulation of the immune system [27,46,290,291,292,293].

However, persistently high levels of PRL can lead to (secondary to GnRH inhibition) hypogonadotropic hypogonadism related to the suppression of both LH and FSH, as well as fertility problems that, in women, are most often a result of ovulation disorders, including the complete disappearance of ovulation [45]. Perhaps surprisingly, only a very small percentage of GnRH neurons express PRLRs or exhibit STAT5 phosphorylation (pSTAT5) in response to an acute PRL stimulus. Moreover, the interaction of PRL with the cell membrane of GnRH neurons is not accompanied by significant modulation of its excitability [294,295]. In addition to changes in PRLR expression under physiological and pathological conditions, the abovementioned findings indicate that PRL does not act directly on GnRH neurons; therefore, the effects of PRL on gonadotropin secretion are mediated by another population of hypothalamic neurons, including those related to kisspeptin secretion [13,296,297].

##### PRL–Kisspeptin Interaction

Kisspeptins are a set of peptide fragments encoded in humans by the KISS1 gene [298]. All of these polypeptides have very similar affinities for the kisspeptin receptor KISS-1R (also known as GPR54), which is a G protein-coupled receptor and is considered the canonical receptor for neuropeptides that are products of the KISS1 gene [299,300].

In the mammalian hypothalamus, kisspeptin neurons are mainly located in two areas: the rostral region, which is associated with the preoptic area (POA), and the caudal region, which is associated with the arcuate nucleus (ARC) [299,301]. It has been proposed that neurons in the ARC that co-express kisspeptin, neurokinin B, and dynorphin (KNDy cells) play key roles in GnRH pulse generation, with kisspeptin driving GnRH release and neurokinin B (NKB) and dynorphin acting as start and stop signals, respectively [301,302]. KNDy neurons in the ARC also appear to mediate the negative feedback effects of E2 and are thought to be the main regulators of pulsatile LH secretion. Moreover, KNDy neurons may also be involved in the positive feedback of E2 to induce the LH surge [303]. However, the role of kisspeptin neurons in the POA has not been determined. KNDy neurons within the POA are hypothesized to be involved in the modulatory effects of E2 on thermoregulation [304].

KISS-1R is expressed by GnRH neurons and is directly activated by kisspeptin to stimulate GnRH release [305]. GnRH-secreting neurons form a relatively small population of cells (e.g., approximately 800 neurons in mice and 1000–2000 in humans) scattered between the POA and ARC in the shape of an “inverted Y” [306].

Kisspeptin is able to induce GnRH secretion both by stimulating KISS-1R in the neuronal stroma and in GnRH nerve terminals located in the mediobasal hypothalamus (MBH) region [307,308]. By acting directly on GnRH release and directly and/or indirectly on LH and FSH secretion, the kisspeptin/KISS-1R system has been widely reported to be a key factor in the regulation of the hypothalamic–pituitary–gonadal (HPG) axis [156,309]. Loss-of-function mutations in the kisspeptin/KISS-1R system disrupt puberty and infertility in both human and animal models, whereas mutations that activate this system lead to precocious puberty in humans [310,311,312,313]. Consistently, it has been hypothesized that abnormal kisspeptin signaling may be responsible for stress-induced fertility disorders, including ovulation disorders [155,156,308,314].

Many studies have shown that the significant actions of PRL in modulating the HPG axis are the result of interactions with neurons expressing the KISS1 gene [13,315,316,317]. This is because most KISS1-expressing neurons co-express functional PRLRs (those receptors through which PRL induces pSTAT5) [318,319]. After acute intraperitoneal (i.p.) administration of PRL to female mice in diestrus, pSTAT induction was observed in approximately 80% of KISS1-expressing neurons in the ARC of the hypothalamus [13,317]. Further confirmation that the PRL/PRL-kisspeptin interaction plays an important role in the regulation of the HPG axis was obtained by demonstrating that systemic or intracerebroventricular (icv) infusion of PRL inhibited KISS1 expression in the hypothalamus, with a subsequent reduction in plasma LH levels [316,320]. Therefore, secondary to the inhibition of GnRH secretion, hypogonadotropic hypogonadism due to hyperprolactinemia is the result of the downregulation of the kisspeptin/KISS-1R signaling system [158,321].

Interestingly, it has been suggested that kisspeptin may have a direct effect on PRL production in lactotrophs of the adenohypophysis under the influence of E2, which may regulate KISS-1R expression and function [322]. KISS-1R appears to be essential for mediating the effect of kisspeptin on PRL secretion, although TIDA neurons do not express KISS-1R and are electrically unresponsive to kisspeptin. It follows that kisspeptin can directly stimulate PRL secretion via KISS-1R in nondopaminergic neurons, whereas the effect on expression of tyrosine hydroxylase TIDA neurons causing an increase in PRL secretion is indirect, which results from inhibition of dopamine release with subsequent pituitary lactotrophs disinhibition. The latter effect doubles the well-documented inhibitory effect of gamma-aminobutyric acid (GABA) within TIDA neurons on dopamine secretion [323]. The kisspeptin-dependent inhibition of dopamine release from TIDA neurons is most likely mediated by the neuropeptide FF receptor 1 (NPFFR1), which is differentially expressed on dopaminergic neurons in the hypothalamus [324]. The occurrence of increased PRL secretion after kisspeptin treatment of TIDA neurons is also dependent on E2 and the expression of estrogen receptor-α (ER-α) on dopaminergic neurons [325].

Kisspeptin, therefore, helps to reverse the suppressive effects of hyperprolactinemia on GnRH neurons. It has been demonstrated that the administration of kisspeptin in hyperprolactinemia induces hypothalamic pulsatile GnRH secretion with subsequent pulsatile LH secretion from the adenohypophysis [326]. Clinical algorithms for kisspeptin treatment in ovulation disorders are constantly being developed due to the beneficial effect of this neuropeptide on follicle maturation, as assessed by maturity-related gene expression [327]. Therefore, the variability of the functional balance within the PRL-kisspeptin interaction may determine the dominant influence of GnRH or PRL on a given stage of the sexual cycle, including ovulation [13,328,329,330].

The Interplay between PRL and kisspeptin in influencing the sexual cycle and ovulation via the HPG axis is shown in Figure 7.

## 4. Concluding Remarks

PRL is a pleiotropic neuroendocrine hormone that is synthesized and secreted mainly by lactotroph cells in the anterior pituitary gland. The principal role of PRL in mammals is to regulate lactation. Although the pool of PRL circulating in the blood is almost exclusively of pituitary origin, one should not forget about the extrapituitary sites of PRL synthesis, which may play a role in local regulation. One such location is the ovarian tissue, where depending on the variable expression of PRLRs during the menstrual cycle, PRL (which acts both in an endocrine and paracrine manner) can significantly influence ovulation and the function of the corpus luteum. However, the nature of these processes corresponding to the physiological state and ovulatory disorders in vivo has still been insufficiently explained, and the results are often contradictory. Nevertheless, the PRL-dependent granulosa cell apoptosis may constitute a future therapeutic target in the treatment of anovulation and/or corpus luteum dysfunction.

Moreover, PRL binding activates PRLR, which is a type-1 family cytokine receptor, which then stimulates a signaling cascade through the activation of STAT5. Thus, in patients with hyperprolactinemia, a shift in the immune system balance between proinflammatory and anti-inflammatory factors may manifest itself at the ovarian level as an ovulation disorder (e.g., resulting from abnormal rupture of the wall of the Graafian follicle). In this situation, attempts to influence PRLR expression/signaling via PRLRs in ovarian follicle tissue may prove promising.

Ovulation disorders involving more or less severe symptoms of hypogonadotropic hypogonadism accompanied by hyperprolactinemia absolutely require diagnostic methods to exclude/confirm prolactinoma and side effects of the utilized drugs (iatrogenic causes) and, subsequently, less common causes of elevated PRL. 

Finally, an important consideration in the clinical approach to treating hyperprolactinemia is macroprolactinemia. Due to the fact that macroprolactin interferes with many immunological assays that are commonly used for the detection of PRL, macroprolactinemia is a frequent cause of misdiagnosed hyperprolactinemia in clinical practice for the treatment of ovulation disorders. Patients with undiagnosed macroprolactinemia may then be unnecessarily exposed to pituitary imaging and to futile treatment with anti-PRL drugs (DA agonists), to which they are resistant. However, due to the fact that the main cause of macroprolactinemia involves anti-PRL antibodies, the diagnosis of a patient with infertility and ovulation disorders should include accompanying autoimmune diseases.

## Figures and Tables

**Figure 1 ijms-25-01976-f001:**
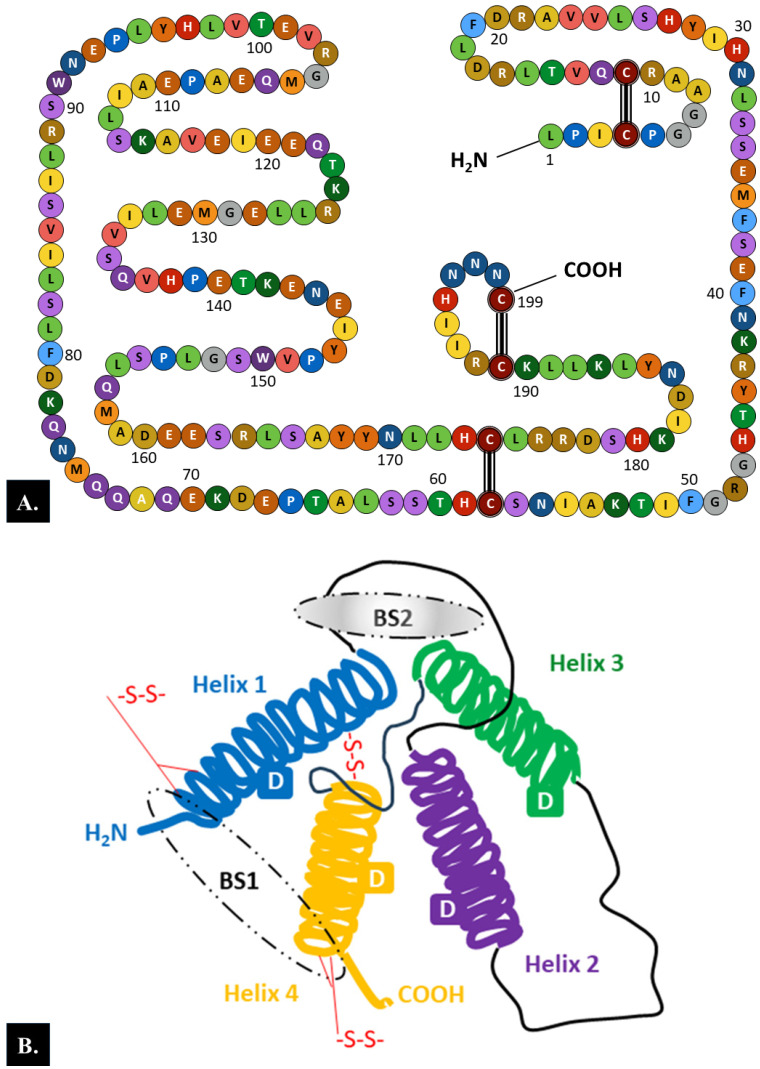
Structure of the prolactin (PRL) molecule. (**A**). **The 199 amino acid chain**. There are three intramolecular disulfide bonds between six cysteine residues in the human prolactin chain: Cys4–Cys11, Cys58–Cys174, and Cys191–Cys199. (**B**). **The tertiary structure**. Each PRL molecule contains four α-helical domains (D) and two binding sites (BSs) (BS1 involves helices 1 and 4, while BS2 encompasses helices 1 and 3) [6,47]. Loops between the helices and three intermolecular disulfide bonds between six cysteine residues are also marked.

**Figure 2 ijms-25-01976-f002:**
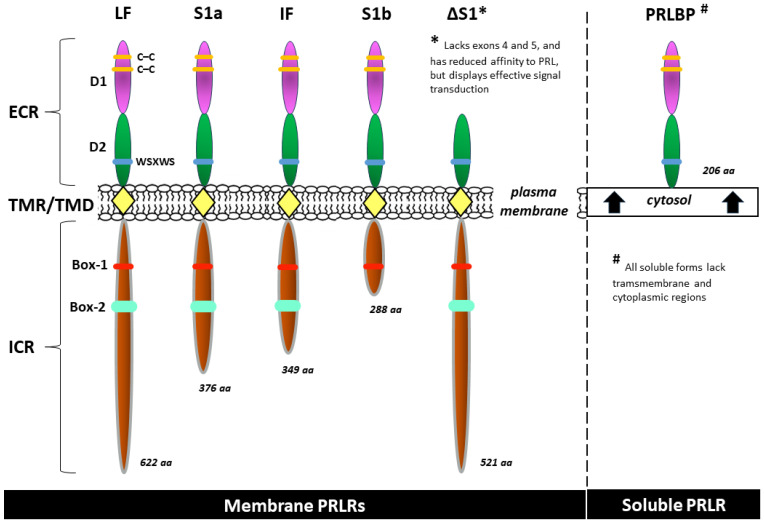
Examples of common isoforms of the human prolactin receptor (PRLR): long form (LF), intermediate form (IF), short isoforms (S1a, S1b), isoform ΔS1, and, as a representative of the soluble forms, the prolactin binding protein (PRLBP) identified in human serum and milk [62,64,69,70,72]. aa—amino acid count; Box-1 and Box-2—the proline-rich and hydrophobic regions in the intracellular domain of cytokine receptor 1 and 2, respectively; C–C—carbon–carbon bond; D1, D2—the two fibronectin type III domains; ECR—extracellular region; ICR—intracellular (cytoplasmic) region; TMR/TMD—transmembrane region/transmembrane domain; WSXWS—a conserved amino acid sequence (WS motif) serving as a molecular switch involved in PRLR activation.

**Figure 3 ijms-25-01976-f003:**
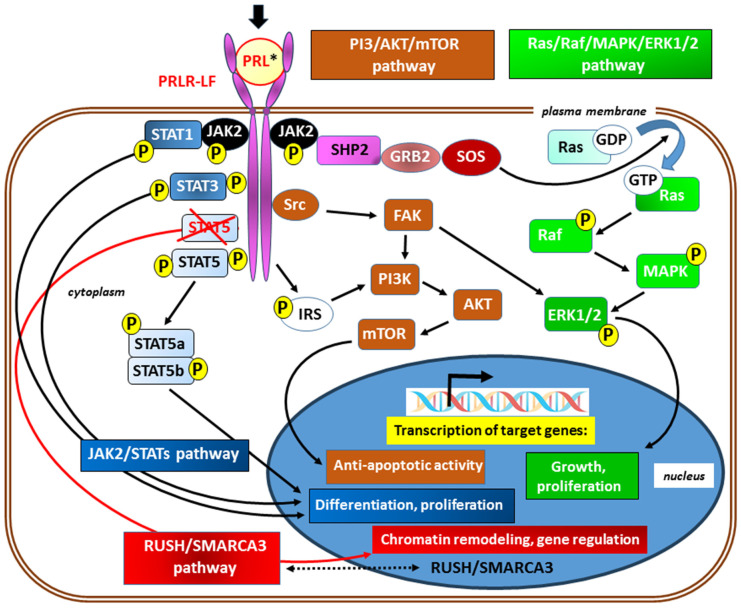
Simplified scheme of signaling via the prolactin receptor; its long isoform (PRLR-LF) has been included for better clarity. The main signaling pathways and the typical known effects of their activation are marked in different colors. In canonical (JAK/STAT) signaling (shades of blue), ligand binding to the PRLR (black arrow) results in stimulation of the tyrosine kinase activity of Janus kinase 2 (JAK2) and subsequent multiple tyrosine residues phosphorylation (P) and activation of signal transducer and activator of transcription (STATs) proteins, in particular, STAT5 (JAK2/STAT5 signaling) [75,76]. The lack of a physical association with STAT5a, as a result of STAT5 blocking (marked with crossed red lines), may redirect to signaling via RUSH/SMARCA3 (*red*) with gene regulation mainly through chromatin remodeling. The presence of transcription factors RUSH-1α and RUSH-1β, as well as SWI/SNF-related matrix-associated actin-dependent regulator of chromatin subfamily A, member 3 (SMARCA3) in both cytoplasm and nucleus, suggests their bidirectional passage across the nuclear membrane (marked with a double-sided arrow with a dotted line) [78]. Transcription of target genes also occurs after activation of the Ras kinases/Raf kinases/mitogen-activated protein kinase/extracellular signal-regulated kinase ½ (Ras/Raf/MAPK/ERK1/2) [77,78] and the phosphoinositide 3-kinase/protein kinase B/mammalian target of rapamycin (PI3-Kinase/AKT/mTOR) [78,79] pathways. These signal paths are integrated with each other on many levels. * Activation of the PRLR may occur due to the action of agonists other than prolactin (e.g., hPL, GH, and IL-2). Other abbreviations: FAK—focal adhesion kinase; GDP and GTP—guanosine diphosphate and guanosine triphosphate, respectively; GRB2—growth factor receptor-bound protein 2; IRS—insulin receptor substrate; SHP2—a Src homology 2 (SH2) domain-containing non-transmembrane protein tyrosine phosphatase; SOS—son of sevenless; refers to a set of genes encoding guanine nucleotide exchange factors that act on the Ras subfamily of small GTPases; Src—a non-receptor cytoplasmic tyrosine kinase.

**Figure 4 ijms-25-01976-f004:**
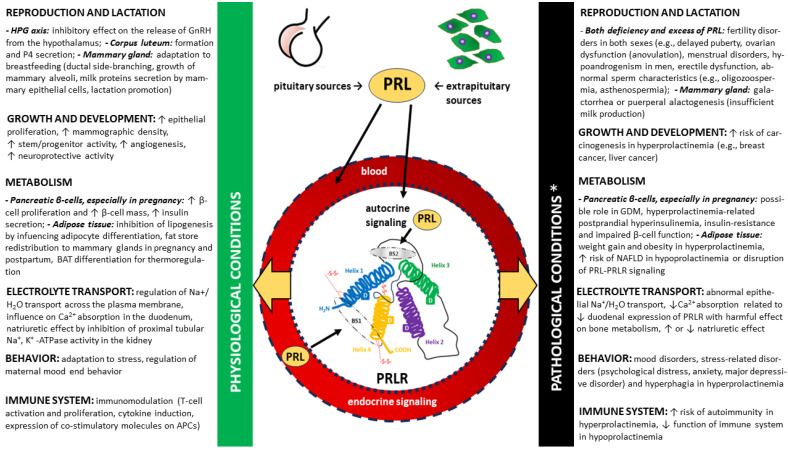
A simplified overview of the diverse functions of the prolactin receptor (PRLR) signaling through endocrine and autocrine prolactin (PRL) binding in normal and pathological conditions (marked with yellow arrows, respectively) related to the functioning of the hypothalamic–pituitary–gonadal (HPG) axis [7,44,46,80,81,82,83], growth and development [84,85,86,87,88,89,90,91,92], metabolism [17,20,33,93,94,95,96,97,98,99,100,101,102,103,104,105], electrolyte transport [106,107,108,109], behavior [24,110,111,112,113] and immune system [25,26,27,33]. ↑—increase, ↓—decrease, APC—antigen-presenting cells, BAT—brown adipose tissue, GDM—gestational diabetes mellitus, GnRH—gonadotropin-releasing hormone (gonadoliberin), NAFLD—non-alcoholic fatty liver disease, P4—progesterone. * Associated with both hyper- and hypoprolactinemia and disorders of PRLR expression/function. In contrast to hyperprolactinemia, the most common cause of which is a prolactinoma, in the vast majority of cases of hypoprolactinemia, the deficiency occurs secondary to general anterior pituitary dysfunction, as is the case in Sheenan’s syndrome (a postpartum necrosis of the pituitary gland). Isolated hypoprolactinemia is rare, and it may have a genetic component (i.e., familial puerperal alactogenesis).

**Figure 5 ijms-25-01976-f005:**
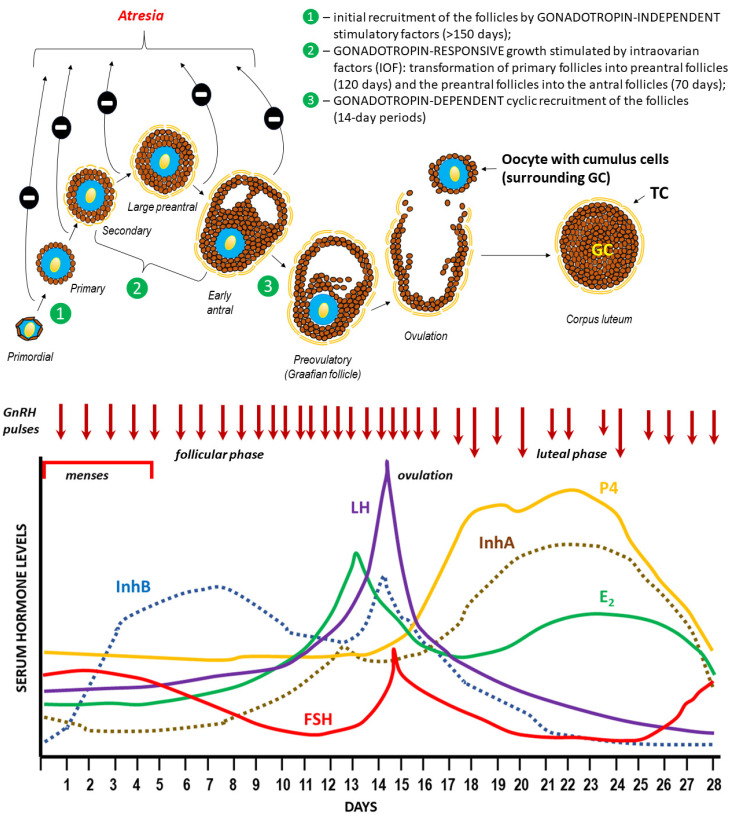
Folliculogenesis, ovulation, and hormonal regulation of the menstrual cycle [115,116,117,118,119,120,121,122,123,124].

**Figure 6 ijms-25-01976-f006:**
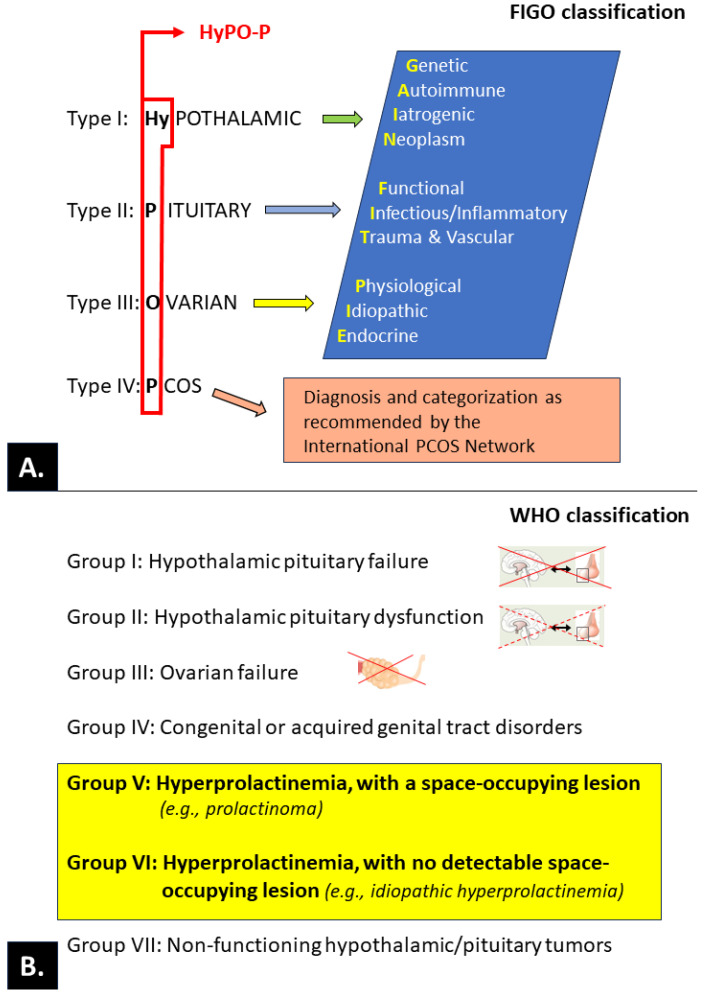
Classification of ovulatory disorders according to the International Federation of Gynecology and Obstetrics—FIGO (**A**) [162] and the World Health Organization—WHO (**B**) [163,164]. The primary level of the **FIGO system** is based on an anatomic model (hypothalamus, pituitary, ovary) that is completed with a separate category for polycystic ovary syndrome (PCOS; the acronym HyPO-P). Each anatomic category is stratified in the second layer of the system to provide granularity for investigators, clinicians, and trainees using the “GAIN-FIT-PIE” mnemonic (genetic, autoimmune, iatrogenic, neoplasm; functional, infectious and inflammatory, trauma and vascular; physiological, idiopathic, endocrine). The tertiary level allows for specific diagnostic entities, including those related to PCOS.

**Figure 7 ijms-25-01976-f007:**
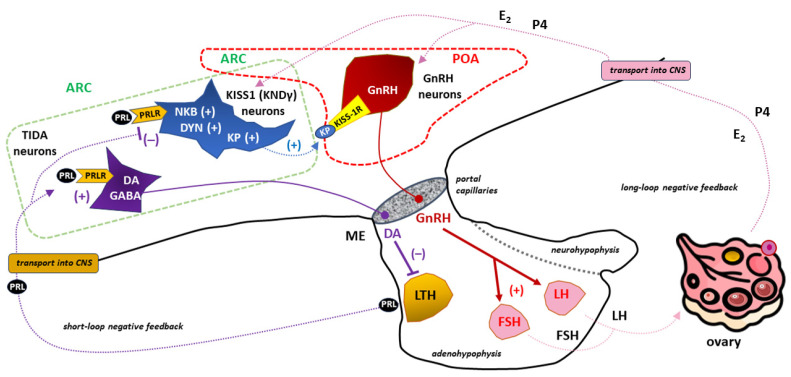
The importance of (tuberoinfundibular dopamine) TIDA, (kisspeptin/neurokinin B/dynorphin (KNDγ or KISS1), and gonadotropin-releasing hormone (GnRH)-secreting neurons in the prolactin (PRL)- and kisspeptin (KP)-dependent regulation of the hypothalamic–pituitary–gonadal (HPG) axis in females. TIDA neurons located in the arcuate nucleus (ARC; marked with a dashed green line) send axons towards the median eminence (ME). The tonic release of dopamine (DA) from these neuronal endings provides an appropriate level of inhibition of PRL release from lactotrophs (LTH) under physiological conditions [8]. PRLR expression in TIDA neurons is crucial for the operation of such a short negative feedback loop in the DA/PRL system [270,271,272]. PRL in a sufficiently high concentration causes suppression of GnRH neurons that are scattered between the preoptic area (POA; marked with a dashed red line) and ARC in the shape of an “inverted Y” [82]. Due to the negligible expression of PRLR in GnRH neurons, the inhibitory effect of PRL is exerted indirectly through the participation of KNDy neurons located within the ARC [13,294,295,296,297]. PRLR stimulation of KNDy neurons leads to inhibition of KP secretion, a family of polypeptides that stimulate GnRH secretion via the KP receptor (KISS-1R, also known as GPR54) in GnRH neurons [315,316,317]. PRL-dependent, limited stimulation of GnRH neurons with KP to release GnRH translates into reduced release of gonadotropins from the anterior pituitary gland with the subsequent manifestation of hypogonadotropic hypogonadism [45,314]. Hormones of ovarian origin, mainly estradiol (E_2_) and progesterone (P4), operating in a long negative feedback loop, influence the activity of GnRH neurons with a force proportional to their concentration in the blood. Other abbreviations: CNS—central nervous system, DYN—dynorphin, FSH—follicle-stimulating hormone, GABA—gamma-aminobutyric acid, LH—luteinizing hormone, NKB—neurokinin B, “+”—stimulation, “−”—inhibition.

**Table 3 ijms-25-01976-t003:** Causes of hyperprolactinemia and potentially ovulatory disorders.

Category of Hyperprolactinemia	Examples, Incl. Medications	Possible Etiology (Direct Cause, If Known)
PHYSIOLOGICAL	Pregnancy and postpartum Breast feeding (nipple stimulation, suckling) Nursing	Reduced dopamine secretion from tuberoinfundibular dopamine (TIDA) neurons of the arcuate nucleus in the hypothalamus and disappearance of the negative-feedback mechanism between PRL and TIDA activity, resulting in unresponsiveness of TIDA to increased PRL concentration [232].
	Stress	Complex neuroendocrine response to physical and emotional distress reduces PRL signaling in TIDA neurons and thus potentially a decline in their inhibitory dopamine-dependent influence on PRL secretion [233].
	Exercise, especially high intensity	Hypothalamic thyrotropin-releasing hormone (TRH) release caused by physical exercise stimulates both release of thyroid-stimulating hormone (TSH) and thus the production of thyroid hormones as well as the release of PRL from lactotrophic cells [234].
	Sleep	Unspecified connection with the sleep-wake cycle [235].
	Chest wall stimulation	Neurogenic. It is hypothesized that even indirect stimulation of the intercostal nerves may result in induction of a stimulatory reflex conducted by intercostal nerves with subsequent stimulation of hypothalamic centers controlling lactation [236].
	Food ingestion	Probably because PRL signaling is implicated in the regulation of glucose homeostatic adaptations through its impact in pancreatic islet cell physiology and glucose metabolism [237].
	Sexual intercourse	PRL increases following orgasm are involved in a feedback loop that serves to decrease arousal through inhibitory central dopaminergic and probably peripheral processes [238].
PATHOLOGICAL	Pituitary disease: – Pituitary tumors: prolactinomas, mixed GH/PRL or adrenocorticotropic hormone (ACTH)/PRL—secreting adenomas, intrasellar non-secretory tumors causing stalk compression (non-functioning adenomas, germinoma, meningioma, glioma, metastasis) – Hypophyseal stalk lesion (“stalk effect”) – Hypophysitis (inflammation of hypophyseal stalk, e.g., lymphocytic hypophysitis) primary or secondary to a local or systemic process – Hypothalamic and pituitary stalk disease: granulomatous disease (sarcoidosis, tuberculosis, eosinophilic granuloma), tumors (craniopharyngioma, hamartoma, glioma, germinoma, metastasis), cranial irradiation, pituitary stalk section, empty sella syndrome (empty pituitary fossa), vascular (aneurysm, arteriovenous malformation)	Differential diagnosis is broad (including primary tumors, metastases, and lympho-proliferative diseases) and multifaceted. Excessive production of PRL by hypertrophic or neoplastic pituitary tissue and/or disinhibition of lactotrophs due to mechanical interruption of the portal transport of dopamine from the hypothalamus to the anterior pituitary gland (stalk effect) [82,239].
	Primary hypothyroidism	Increased levels of TRH can cause to rise PRL levels by stimulation of TRH receptors on lactotrophic cells [240].
	Chronic renal insufficiency	Reduction in metabolic clearance of PRL and direct stimulation of PRL secretion from lactotrophs due to reduced availability of dopamine in the brain in the uremic state [241,242].
	Severe liver failure (inc. cirrhosis)	Decompensated liver function leads to an alteration in the type of amino acids entering the central nervous system with an increase in the synthesis of false neurotransmitters such as octopamine and phenylethanolamine. These false neurotransmitters may inhibit the dopamine release contributing to hyperprolactinemia [243].
	Neuraxis irradiation (radiation therapy)	Hyperprolactinemia may develop in 20–50% of cases after high dose (>50 Gy) cranial radiotherapy as a result of hypothalamic damage and reduced inhibitory dopamine activity [244]. Elevated PRL levels may decline and normalize during follow-up due to radiation-induced reduction of the pituitary lactotroph cells [245].
	Spinal cord lesions	Secondary hyperprolactinemia due to elevated endorphins and opioid phenotypes in the central nervous system (CNS) following spinal cord injury-related shock as well as production of PRL-releasing factors [246,247,248].
	Seizures	Activation of certain pathways in the brain that regulate PRL secretion or the activation of the hypothalamic–pituitary–adrenal (HPA) axis, which can subsequently increase PRL levels in the blood [249].
	Polycystic ovary syndrome (PCOS)	Probably secondary to increased E_2_ levels and/or insulin-resistance in PCOS patients [224,250].
	Ectopic secretion of PRL (bronchogenic carcinoma, hypernephroma)	Excess PRL from tumor tissue not subject to negative feedback regulation by dopamine [251].
	Chest wall trauma (including surgery, herpes zoster)	A damage or sectioning of the intercostal nerves may result in reflex stimulation of hypothalamic centers controlling lactation through the same neural pathways involved in puerperal lactation [252].
	Idiopathic	Unknown cause or no clear cause
PHARMACOLOGICAL (iatrogenic)	**Typical antipsychotics:** haloperidol, phenothiazines, thioridazine, clomipramine, fluphenazine, pimozide, prochlorperazine **PRL-raising atypical antipsychotics:** risperidone, olanzapine, molindone, paliperidone	Antipsychotic-associated dopamine D_2_ receptor antagonism. Blockade of D_2_ by typical antipsychotics and risperidone can cause hyperprolactinemia [253,254]. Atypical antipsychotics other than risperidone are less likely to cause sustained hyperprolactinemia; asymptomatic and transient hyperprolactinemia is more common, because of their lower affinity for D_2_ receptors [255].
	**Antidepressant agents:** selective serotonin reuptake inhibitors—SSRIs (citalopram, escitalopram, fluoxetine, fluvoxamine, milnacipran, paroxetine, sertraline, venlafaxine); tricyclic antidepressants—TCAs (amitriptyline, amoxapine, clomipramine, desipramine); monoamine oxidase inhibitors—MAOIs (clorgyline, pargyline)	Inhibition of the tuberoinfundibular dopaminergic (D_2_) pathway through stimulation of gamma aminobutyric acid (GABA)ergic neurons and release of PRL-regulating factors, such as vasoactive intestinal peptide (VIP) or oxytocin [256,257]. May cause indirect modulation of PRL release by increasing serotonin (5-HT) [258].
	**Gastrointestinal drugs:** metoclopramide, domperidone, prochlorperazine, metiamide, cimetidine (intravenous)	Dopamine D_2_ receptor antagonism [259]. Cimetidine, a histamine H_2_ receptor antagonist acting in the hypothalamus inhibits dopamine secretion, as well as increases 5-HT release within dopamine-GABA-serotoninergic system [258,260,261].
	**Antihypertensive agents:** - methyldopa, - verapamil, - reserpine, tetrabenazine	Alpha-methyldopa causes hyperprolactinemia by inhibiting the enzyme l-aromatic amino acid decarboxylase (which is responsible for converting L-dopa to dopamine) and by acting as a false neurotransmitter to decrease dopamine synthesis [262]. Verapamil, a phenylalkylamine calcium channel blocker, blocks hypothalamic (tuberoinfundibular) production of dopamine [263]. Reserpine and tetrabenazine produce a reversible depletion of dopamine by inhibition of the vesicular monoamine transporter type-2 (VMAT2) that blocks dopamine storage in synaptic vesicles of neurons [264,265].
	**Opiates:** codeine, morphine	Disinhibition of lactotrophs by the inhibitory effect of TIDA neurons (decreased dopaminergic activity due to decrease in the turnover and release of hypothalamic dopamine) [266,267].
	**Hormone preparations:** antiandrogens, combined oral contraceptives, estrogens	Estrogen-stimulated lactotroph hyperplasia [268,269].
ANALYTICAL (assay artefacts)—misdiagnosis	Macroprolactin	Macroprolactinemia represents a state of hyperprolactinemia characterized by the predominance of macroprolactin (also known as big–big PRL), a macromolecule with limited bioavailability and bioactivity, and it is mainly suspected in asymptomatic individuals or those without the typical hyperprolactinemia-related symptoms [54,57,231].
	Heterophilic antibodies (endogenous proteins that bind animal antigens)	Heterophilic antibodies interact poorly and nonspecifically with the assay antibodies. Depending on the type of such antibody and the immunoassay format, heterophilic antibodies can lead to both falsely high and low analyte concentrations according to the site of interference [231].
	Prozone phenomenon (hook effect), i.e., falsely normal or mildly elevated PRL while the true PRL concentration is many fold higher than the upper limit	This phenomenon occurs when extremely high PRL concentration (i.e., observed in large pituitary macroadenomas (≥3 cm)) saturates both the capture and the labeled antibody during immunoassay, preventing the formation of the “sandwich” and causing false-negative results [231].

## Data Availability

No new data were created. Instead, the data are quoted from the available cited literature.

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
