# Peer review of "Current Insights in Prolactin Signaling and Ovulatory Function"

_ijms, 2024, doi:10.3390/ijms25041976_

Round 1
Reviewer 1 Report
Comments and Suggestions for Authors
Please see the attachment.

English is good enough to be understandable.
Reviewer 2 Report
Comments and Suggestions for Authors
The review is well organized and very detailed. Can the authors include more novel findings in this field and make the session 2, session 3.1 and 3.2 more concise? Reviews of science should be different from textbook.
- The author listed the hormones regualting menstrual cycle in a very detailed table (Table 2). Can the author include the effect of prolactin on menstrual cycle regulation? If there is no evidence that prolactin has independent effect on mestrual cycle, Table 2 can be reducted.
- Table 3 listed causes of hyperprolactinemia. Do these causes of hyperprolactinemian also induce ovuolation disorder? If so, it will better to integrate these causes with ovulation disorder in this review and describe the association betweeen them.
- I highlighte some spelling mistakes:
In Figure 4,
BEHAVIOR session: bebawior
IMMUNE SYSTEM session: co-stymulatory moleculaes
METABOLISM session, hyperprolactynemia. What is NAFLD in this session?
- In session 3 Ovulation, paragraph 3, the author mentioned “However, such assumptions have not been confirmed under in vitro conditions, wherein apoptosis of human granulosa cells (hGLC) and transfected cell lines is induced by high doses of FSH or FSHR overexpression, whereas estrogens induce antiapoptotic signals via nuclear and a G protein-coupled estrogen receptor (GPER).” What gene was transfected into the cell lines?
- In session 3 Ovulation, paragraph 6, this sentence may need to be rewritten, “Thus, out of the entire pool of ovarian follicles that are present on the day of the woman’s birth, ovulation will increase in only 0.1% of patients, whereas ovulation will increase in 99.9% of patients throughout the process of follicular atresia.”
Round 2
Reviewer 1 Report
Comments and Suggestions for Authors
-
Reviewer 2 Report
Comments and Suggestions for Authors
The manuscript was sufficiently improved to warrant publication.